# Clinical and Radiological Outcomes for Guided Implant Placement in Sites Preserved with Bioactive Glass Bone Graft after Tooth Extraction: A Controlled Clinical Trial

**DOI:** 10.3390/biomimetics7020043

**Published:** 2022-04-13

**Authors:** Priyanka Baskaran, P.S.G. Prakash, Devapriya Appukuttan, Maryam H. Mugri, Mohammed Sayed, Sangeetha Subramanian, Mohammed Hussain Dafer Al Wadei, Zeeshan Heera Ahmed, Harisha Dewan, Amit Porwal, Thodur Madapusi Balaji, Saranya Varadarajan, Artak Heboyan, Gustavo V. O. Fernandes, Shankargouda Patil

**Affiliations:** 1Department of Periodontology and Oral Implantology, SRM Dental College, Ramapuram Campus, Chennai 600089, India; pbaskaran94@gmail.com (P.B.); devapriyamds@gmail.com (D.A.); sangeetha_doc@yahoo.com (S.S.); 2Department of Maxillofacial Surgery and Diagnostic Sciences, College of Dentistry, Jazan University, Jazan 45412, Saudi Arabia; dr.mugri@gmail.com; 3Department of Prosthetic Dental Sciences, College of Dentistry, Jazan University, Jazan 45412, Saudi Arabia; drsayed203@gmail.com (M.S.); harisha.dewan@yahoo.com (H.D.); aporwal2000@gmail.com (A.P.); 4Department of Restorative Dental Science, King Khalid University, Abha 61421, Saudi Arabia; moalwadai@kku.edu.sa; 5Department of Restorative Dental Sciences, College of Dentistry, King Saud University, Riyadh 11451, Saudi Arabia; aheera@ksu.edu.sa; 6Tagore Dental College and Hospital, Chennai 600089, India; tmbala81@gmail.com; 7Department of Oral Pathology and Microbiology, Sri Venkateswara Dental College and Hospital, Chennai 600089, India; vsaranya87@gmail.com; 8Department of Prosthodontics, Faculty of Stomatology, Yerevan State Medical University after Mkhitar Heratsi, Str. Koryun 2, Yerevan 0025, Armenia; heboyan.artak@gmail.com; 9Periodontics and Oral Medicine Department, University of Michigan School of Dentistry, Ann Arbor, MI 48109, USA; 10Department of Maxillofacial Surgery and Diagnostic Sciences, Division of Oral Pathology, College of Dentistry, Jazan University, Jazan 45142, Saudi Arabia

**Keywords:** bioactive glass, CBCT, dental implants, flapless technique guided implant surgery, implant planning, minimally invasive surgical technique, surgical guides

## Abstract

The goal of the study was to evaluate marginal bone loss (MBL) after 1-year implant placement using a guided implant surgical (GIS) protocol in grafted sockets compared to non-grafted sites. We followed a parallel study design with patients divided into two groups: grafted group (Test group, *n* = 10) and non-grafted group (Control, *n* = 10). A bioactive glass bone graft was used for grafting. A single edentulous site with a minimum bone height ≥11 mm and bone width ≥6 mm confirmed by cone-beam computerized tomography (CBCT) was chosen for implant placement. Tapered hybrid implants that were sandblasted and acid-etched (HSA) were placed using the GIS protocol and immediately loaded with a provisional prosthesis. MBL and implant survival rates (ISR) were assessed based on standardized radiographs and clinical exams. Patients were followed up for 1-year post-loading. MBL after one year, in the control group, was −0.31 ± 0.11 mm (mesial) and −0.28 ± 0.09 mm (distal); and in the test group was −0.35 ± 0.11 mm (mesial) and −0.33 ± 0.13 mm (distal), with no statistical significance (*p* > 0.05). ISR was 100% in both groups after one year. ISR was similar between groups and the marginal bone changes were comparable one year after functional loading, without statistical significance, suggesting that bioactive glass permitted adequate bone formation. The GIS protocol avoided raising flaps and provided a better position to place implants, preserving the marginal bone around implants.

## 1. Introduction

The post-extraction remodeling of the alveolar socket can exert horizontal and vertical dimensional changes on the underlying bone [1,2]. The preservation of the extraction socket is vital to prevent potential morbidity and collapse in the implant sites [3]. Various socket preservation techniques, such as bone grafts with or without membranes, have been studied and implemented, demonstrating predictable outcomes by retaining the ridge dimensions for implant fixture placement [4].

Bioactive glass is a time-tested regenerative biomaterial with osteoconductive properties. It has the advantages of enhanced biocompatibility, resorption at the right time, ion leaching properties, and the conduciveness to support the migration and proliferation of osteogenic cells [5]. The grafting material used for socket augmentation prior to implant placement was a synthetic bioactive bone graft material (NovaBone® Dental Morsels, NovaBone Products, Alachua, Fl, USA) with calcium phosphosilicate as the active component. The material is osteostimulatory and the osteoconductive composite of bioceramic containing oxides of silicon, calcium, sodium, and phosphorous [6]. The material, once placed into the socket, comes into contact with the aqueous body fluid initiating chemical reaction at the surface of the graft. There is a release of sodium, silica, calcium, and phosphate ions. The silica released forms silanol groups (Si-OH) on the surface which repolymerize into a silica-rich layer; simultaneously, there is the precipitation of amorphous calcium phosphates which undergo crystallization into hydroxyl carbonate apatite. This stimulates osteoblast recruitment, proliferation, and differentiation. Furthermore, continuous ion release transforms the graft material into a porous scaffold that promotes new bone formation [7,8,9,10].

Fiorelliniet al. [11], based on their systematic review, suggested that implants placed in grafted edentulous sites using the conventional placement technique had an implant survival rate (ISR) comparable to implants placed on a native bone (without bone grafts). Similarly, Urban et al. [12] observed a 100% cumulative implant survival rate at six years post-loading in thirty-six previously grafted sites.

The quality and quantity of the available alveolar bone determine the long-term stability of implants. In recent times, the outcome of therapy has been evaluated based on implant survival and minimal marginal bone loss (MBL) post-functional loading. Barone et al. [13] and Simion et al. [14] suggested that sockets that received bone grafts were restored with implants that had greater marginal bone loss.

Standardizing implant placement can overcome the drawbacks of conventional surgeries, such as improper angulation and the incorrect positioning of dental implants. Extensive flap elevation can decrease the supra-periosteal blood supply [15]. Guided implant surgery could offer a solution to these challenges. GIS is a process of planning the implant surgery based on CBCT and computer-aided design (CAD)/computer-assisted navigation technology. It is a routine clinical procedure with a predictable outcome and high success rate.

A fully guided surgical implant protocol employs a surgical guide to assist the clinician, from the initial step (osteotomy) to implant placement, using driving sleeves innately present within the guide. Surgical guides provide higher predictability and accuracy in transferring the virtual implant position to the patient’s mouth compared to half-guided surgery [16]. A major advantage of the GIS technique is that the blood vessels around the implants are preserved, as there is no flap elevation. This helps in reducing MBL. Tallarico et al., observed lower MBL rates with computer-guided implant placement compared to conventional implant placement [17]. 

Reducing MBL rates is an important factor for implant survival. The use of GIS could result in reduced MBL. This study aimed to evaluate and compare the MBL and implant survival rates (ISR) in graft sites with a bioactive glass, one year after implant placement, using a GIS protocol with immediate functional loading compared to sites without grafting.

## 2. Materials and Methods

### 2.1. Trial Design

The study was conducted after receiving ethical clearance from the Institutional Review Board (SRMDC/IRB/2018/MDS/No.502) SRM Dental College, Chennai, India. This study followed a prospective controlled clinical trial (CCT) design based on a cohort of consecutive patients with an allocation ratio of 1:1. Patients were recruited from the out-patient clinics of the Department of Periodontology, SRM Dental College. Written informed consent was obtained from all participants after a thorough explanation of the aims and objectives of the study. All interventions were in accordance with the ethical standards of the revised Helsinki Declaration for Biomedical Research involving human subjects. This clinical trial was registered at Clinicaltrials.gov (accessed on 19 February 2022) (ID:CTRI/2019/09/021168). The present study was reported based on the CONSORT statement [18], and followed the EQUATOR guidelines.

### 2.2. Sample Size Calculation 

A sample size calculation was done based on results from an earlier study by Pozzi et al. [19], with 5% alpha error and 80% power based on MBL measurements, with ten implants per group for a total of 20 implants.

### 2.3. Inclusion and Exclusion Criteria

Inclusion criteria were: (i) age between 19–50 years; (ii) good oral hygiene and stable periodontal status; (iii) single edentulous sites with healthy teeth on both sides; (iv) minimum bone height ≥11 mm and bone width ≥6 mm during baseline CBCT evaluation; (v) willingness to participate in the study and comply with the necessary study requirements, including follow-up for one year.

Exclusion criteria included (i) patients with a habit of chronic intake of analgesics; (ii) patients under treatment with bisphosphonates and corticosteroids; (iii) current smokers (>10 cigarettes per day); (iv) loss of any bony wall during the extraction or augmentation; (v) extraction sites associated with failed endodontic treatment; (vi) trauma associated fracture and sub-gingivally extending fracture lines that cannot be endodontically restored; (vii) tooth with poor prognosis; (viii) residual root stumps; (ix) pregnant and lactating female patients; (x) untreated periodontitis; (xi) presence of parafunctional habits; (xii) immunocompromised patients; (xiii) active infection or severe inflammation at the site of implant placement, evaluated by the presence of purulent secretion. 

### 2.4. Blinding

The operator and primary outcome assessors were blinded regarding the recruitment and the sites included in the study (both groups). The patients were not blinded to the intervention.

### 2.5. Implants

A total of twenty dental implants (*n* = 20), tapered hybrid sandblasted and acid-etched (HSA, Dio-Navi^®^, Busan, Korea), were placed in patients with single edentulous sites which had previously been subjected to tooth extraction. The sites were clinically divided according to whether they received bioactive glass bone graft (Calcium Phosphosilicate [CPS] morsels (NovaBone^®^ Morsels, Alachua, FL, USA) (test group, *n* = 10) or followed the natural healing process (i.e., non-grafted sites) (control group, *n* = 10). The study protocol flow chart is illustrated in Figure 1.

### 2.6. Interventions

A single experienced surgeon (P.S.G.) performed the surgical procedures. The clinical and radiographic parameters were recorded individually by two different investigators (J.C., A.K.) who were blinded to the recruitment and procedures. The sites chosen for implantation received previously bioactive glass bone grafts or followed natural healing after extraction. After six months, the patients underwent implant placement following a specific protocol.

After administering local anesthesia (2% lidocaine with 1:80,000 adrenaline), the patient was asked to rinse the mouth with 0.12% chlorhexidine solution for 30 s. The surgical guide was placed in position and stabilized over the patient’s teeth. The guide had been prepared earlier using a three-dimensional guided navigation software (CBCT). The GIS kit was used to prepare the osteotomy sites for implant placement. The pilot drill was initially driven through the sleeve present in the surgical guide following the manufacturer’s instructions. Subsequent osteotomies were carried out using the drilling protocol provided by the manufacturer. Once the final osteotomy was carried out, implant insertion was done through the sleeve in the surgical guide with an insertion torque greater than 35 N.cm using an implant driver. 

All the implants were placed equi-crestally. The stability was measured after removing the surgical guide, followed by the placement of a prosthetic abutment, and was restored functionally (Figure 2). The same surgical protocol was observed for both the control and test groups. All patients received postoperative instructions. Patients were instructed to rinse with 0.2% chlorhexidine digluconate twice daily for two weeks. Analgesic medications (Ibuprofen, 500 mg) were prescribed a day thrice for three days. 

### 2.7. Clinical and Radiographic Outcomes

Clinical parameters evaluated were full-mouth plaque scores (F-MPS) [20], full-mouth bleeding scores (F-MBS) [21], site-specific plaque scores (S-SPS), site-specific bleeding scores (S-SBS), peri-implant sulcus depth (P-ISD) (using a plastic probe, Hu-Friedy [PCV11KIT12], with a probing force of 0.2 N to 0.3 N), along with peri-implant abutment attachment level (P-IAL) and position of relative gingival margin (PGM) [22].

A radiographic assessment of mean marginal bone loss (MBL) was performed with radiovisiography [23]. A standardization of the radiographs was performed using a customized silicon bite block to index the dentitions that are fixed with the metal bar of the holding device. Radiographs were obtained using a standardized long cone parallel technique. The digital images obtained were superimposed using SOPRO Imaging (v. 2.40). The dimensions obtained measured the marginal bone loss (MBL) at the mesial and distal aspects.

### 2.8. Follow-Up

Clinical parameters of S-SPS, S-SBS, P-ISD, P-IAL, and PGM were evaluated at six months and 1-year post-functional loading. The MBL and implant survival rate (ISR) were assessed at 1-year post-functional loading (Figure 3a–e and Figure 4a–d).

### 2.9. Statistical Analysis

Data management and analysis were performed using Statistical Package for Social Science (SPSS, v. 17, for Microsoft Windows). An independent *t*-test was applied for the intergroup comparison of F-MPI, F-MBI, P-ISD, PGM, MBL, and ISR. Mann–Whitney test U was applied for an intergroup comparison of S-SPS, and S-SBS. A comparison of all the clinical and radiographic parameters within the groups was analyzed through paired *t*-test and Wilcoxon signed-rank test. Pearson’s correlation coefficient test correlated the clinical and radiographic parameters of both the control and test groups. In all the statistical tests, a *p*-value ≤ 0.05 was considered to be significant.

## 3. Results

Twenty-four edentulous sites requiring implant placement in the anterior or pre-molar sites were selected between February 2019 and April 2019. After phase I therapy, patients willing to participate in the whole study period and fill out the inclusion criteria were categorized into two groups. However, two patients in each group declined the participation, thus resulting in a total of 20 patients, with 10 in each group. The parameters were evaluated at three intervals (baseline, after six months, and after 1-year post functional loading). Uniform sample distribution was evident in both groups based on age and gender distribution, as no statistically significant difference was observed (*p* > 0.05).

Clinical parameters, such as S-SPS and S-SBS in the control and test groups, did not show any statistical differences between time points, indicating a healthy peri-implant tissue during loading and a proper peri-implant maintenance post-loading (*p* > 0.05). In the control group, PISD decreased at one year (3.23 ± 0.26 mm) from baseline (3.44 ± 0.29 mm), and from 6 months (3.40 ± 0.24 mm). Similarly, the relative position of the gingival margin showed statistically significant gains at one year (2.50 ± 0.52 mm) compared to the baseline (3.20 ± 0.63 mm), and at six months (2.90 ± 0.56 mm). It indicates a significant coronal creeping of the peri-implant mucosa at the one year period in the control group (*p* < 0.05) (Table 1).

In the test group, P-ISD significantly decreased at one year (3.27 ± 0.34 mm) post-loading and was considerably lower compared to the baseline (3.45 ± 0.35) and at six months (3.42 ± 0.39 mm). The relative position of the gingival margin (PGM) showed a significant gain at six months and one year compared to the baseline (3.40 ± 0.51 mm). It indicates a coronal creeping of the peri-implant mucosa during the one-year follow-up (*p* < 0.05) (Table 1).

The intergroup comparison of the clinical parameters such as S-SPS, S-SBS, P-ISD, and PGM at baseline, six months, and one year revealed no significant differences (*p* > 0.05). These data suggest that patients from both groups had good oral hygiene habits and maintained plaque control measures. There was decreased plaque accumulation and the subjects were free from active peri-implant diseases, leading to an improved soft tissue attachment level (Table 2).

Marginal bone loss (MBL) was compared between groups at one-year post-loading. Both the groups showed similar MBL at the mesial and distal aspects at 1-year post-loading, i.e., 0.31 mm in the mesial aspect and 0.28 mm in the distal aspect of the control group; and 0.35 mm in the mesial and 0.33 mm in the distal part of the test group, without any statistical significance (*p* > 0.05). This indicates that the bone levels had shifted apically with a minimal and comparable amount of MBL in both groups (Table 3).

When correlating the clinical with radiographic parameters, the results were negatively correlated with S-SPS, S-SBS, and P-ISD with MBL. The results showed a positive correlation between the relative position of the gingival margin and MBL. However, none of these comparisons approached statistical significance (*p* > 0.05) (Table 4).

## 4. Discussion

Dental implants serve to provide predictable long-term restorative solutions for missing teeth. Minimizing damage to the underlying alveolar bone can serve to enhance the success and survival rate of dental implants. Several strategies are used to prevent alveolar bone resorption, including atraumatic flapless tooth extraction [24], bone grafts, membranes, and additional surgical procedures [25]. This study assessed the implant survival rate (ISR) and marginal bone loss (MBL) in mesial and distal sites of dental implants placed with guided implant surgical protocol (GIS) in sockets with bioactive glass grafts compared to non-grafted sites after one year of functional loading. Our findings showed that both groups had similar marginal bone loss in the mesial and distal aspects. Pozziet al. (2014) [19] observed an MBL of 0.80 mm in the conventional group and 0.71 mm in the computer-guided group after one year of loading, which depicted less bone loss for the computer-guided group. Similarly, Tallarico et al. (2018) [17] found that computer-guided implant placement reduced the marginal bone loss compared to the free-hand placement over a five-year follow-up after implant placement. The computer-guided group had significantly lower MBL, with a difference of 0.2 mm at 1-year and 0.4 mm at 5-year post-loading. These findings broadly support our result that GIS may help limit marginal bone loss. The type of surgical procedure may play a large role in bone remodeling. Maintaining the architecture of the tissues (soft and hard) around the dental implant and respecting the dynamism of the bone in the healing/remodeling process are critical drivers for success [26]. Using GIS may help to overcome possible drawbacks of the conventional implant placement by avoiding flap elevation, enabling proper angulation, and the accurate positioning of the implants. This is conducive to immediate functional loading, allowing a better remodeling of the underlying local tissues. This can minimize bone loss and help in long-term stability.

Another variable examined was the implant survival rate (ISR). The recent literature reveals that dental implants have a high implant survival rate (96.4%) at ten years post-placement [27]. In this study, we assessed ISR after one year of post-functional loading. The ISR was similar for both groups, with a 100% survival rate. Tallarico et al. [17] and Pozzi et al. [19] demonstrated similar results, highlighting 100% ISR for the computer-guided group.

Bone remodeling directly affects the primary implant stability. Stability is determined by the quality and quantity of bone, the type of implant (geometry, diameter, length, and surface characteristics), and surgical techniques [28,29]. The biomaterial used in grafting may influence secondary stability. Bone graft materials are involved in peri-implant bone remodeling and osteoconduction [30,31], leading to new bone formation at the implant surface [32]. The dental implant used in this study was the tapered hybrid, sandblasted and acid-etched (HSA). Sandblasting and acid-etching techniques improved the bone-to-implant contact (BIC), positively affecting implant stability and reducing bone loss. A study based on the sandblasting technique demonstrated that it accentuated and stimulated osteoblasts to adhere to the implant surface, thereby increasing the rates of bone formation [33]. Similarly, acid-etched implants have also shown high survival rates up to 92.9% in long-term follow-up studies with 17 years of observation [33]. 

The regenerative material used in the present study was bioactive glass morsels. It has osteoconductive activity and serves as a scaffold. Bioactive glass has a brittle structure and has been termed a supercooled liquid. When bioactive glass morsels contact body fluids, they undergo ionic dissolution and glass degradation. The consequent rise in pH of the local microenvironment favors bone regeneration causing enhanced osteoid formation and mineralization [10]. Our findings may be partly due to the high quality of the implant characteristic, regenerative material used, and GIS protocol. The well-preserved extraction socket underwent natural and favorable healing, offering the best environment for plentiful osteoblasts to help in bone formation. The amount of bone healing was comparable in both groups, indicating that bioactive glass is a promising scaffold material. Jung et al. [34] and Ten Heggeler et al. [35] reported that bone substitutes could limit the alveolar resorption after tooth extraction, but not prevent the alveolar bone’s physiological resorption, especially in molar areas.

Our findings suggest that either grafted and non-grafted groups had a reduced S-SPS and S-SBS, ensuring the presence of suitable connective tissue attachment, providing a low level of gingival inflammation. The progressive reduction of P-ISD and positive coronal shift in the PGM of both groups was evident, establishing a soft tissue barrier with the transmucosal component of the implant. Thus, it enhanced the resistance to probing, showed a better mucosal barrier, maintained a reduced P-ISD, and gained in the relative gingival margin position, leading to a reduced peri-implant attachment loss in both groups.

The limitation of this study was that the sample size was relatively small at 20 sites. GIS is technique-sensitive. Beginners or less experienced clinicians could show operator bias. Future studies with greater sample sizes should analyze the clinical and radiological parameters over a long period to provide a fuller picture of the benefits of the GIS protocol in grafted sockets. Different implant designs and surfaces should also be examined to confirm and validate these findings.

## 5. Conclusions

Within the limitations of this study, the guided implant surgical (GIS) protocol appears to be promising, and provides reliable, predictable treatment results, with a reduction in marginal bone loss and improved implant survival rate. The GIS protocol avoided raising flaps and provided a better position to place implants, preserving the marginal bone around implants. The bioactive glass used for grafting permitted adequate bone formation and is a promising scaffolding material. Future research on GIS can serve to validate these findings and provide optimized, customized treatment plans for patients, limiting human error, and improving treatment outcomes.

## Figures and Tables

**Figure 1 biomimetics-07-00043-f001:**
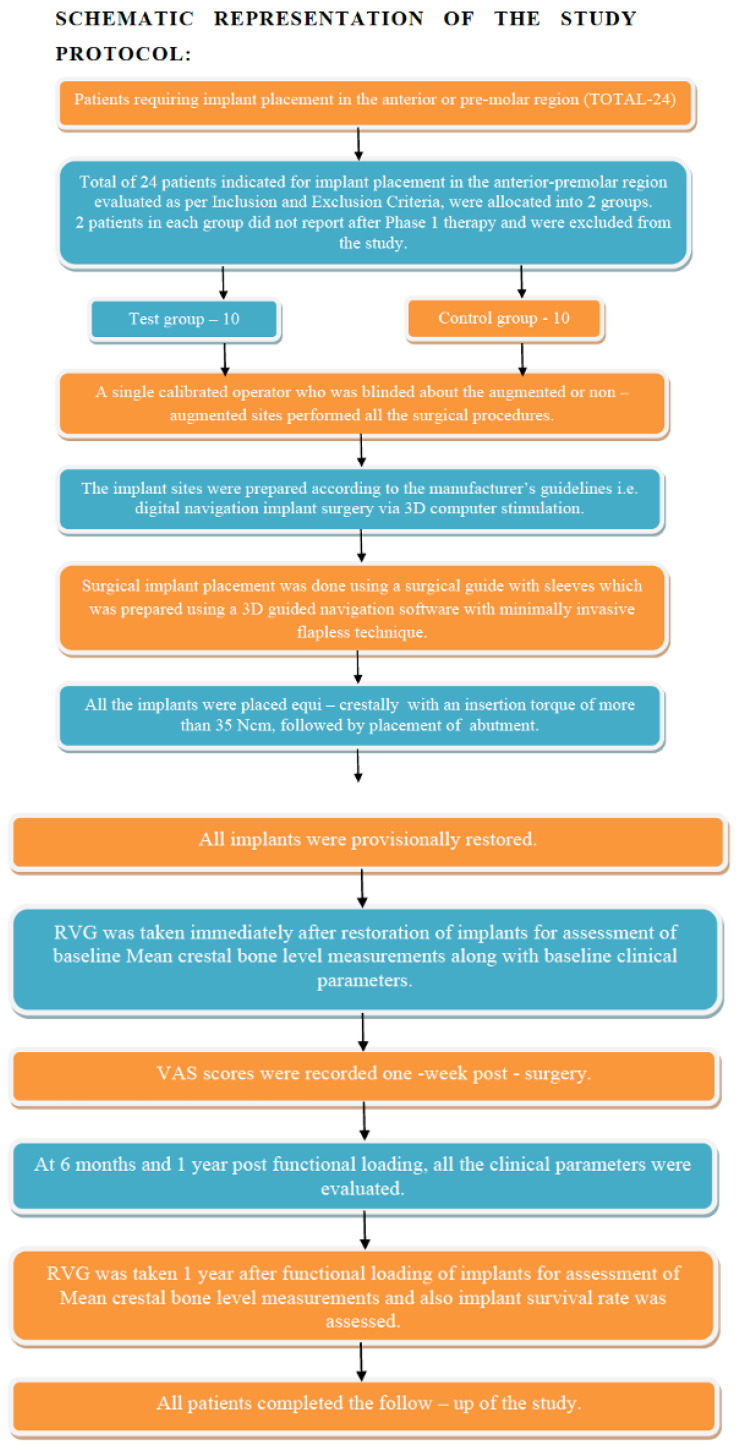
Schematic representation of the study protocol.

**Figure 2 biomimetics-07-00043-f002:**
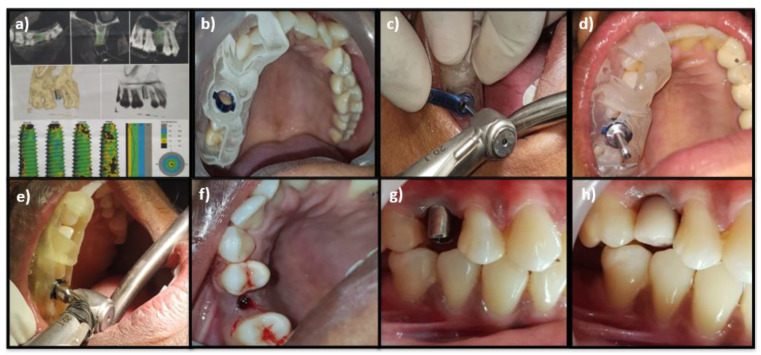
(**a**) Superimposition of CBCT for surgical guide preparation; (**b**) Stabilization of the surgical guide; (**c**) Osteotomy site preparation through the surgical guide using an initial pilot drill; (**d**) Subsequent osteotomies in the sequence of the drilling protocol; (**e**) Implant insertion; (**f**) Implant equi-crestally placed; (**g**) Abutment placed; (**h**) Provisional restoration.

**Figure 3 biomimetics-07-00043-f003:**
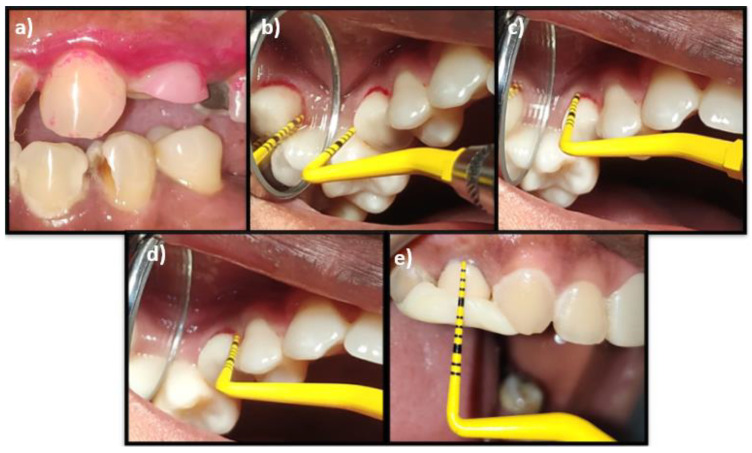
(**a**) Evaluation of site-specific plaque score using a disclosing agent; (**b**) Evaluation of the site-specific bleeding score; (**c**) Measurement of peri-implant sulcus depth; (**d**) Measurement of peri-implant abutment attachment level; (**e**) Measurement of the position of the relative gingival margin.

**Figure 4 biomimetics-07-00043-f004:**
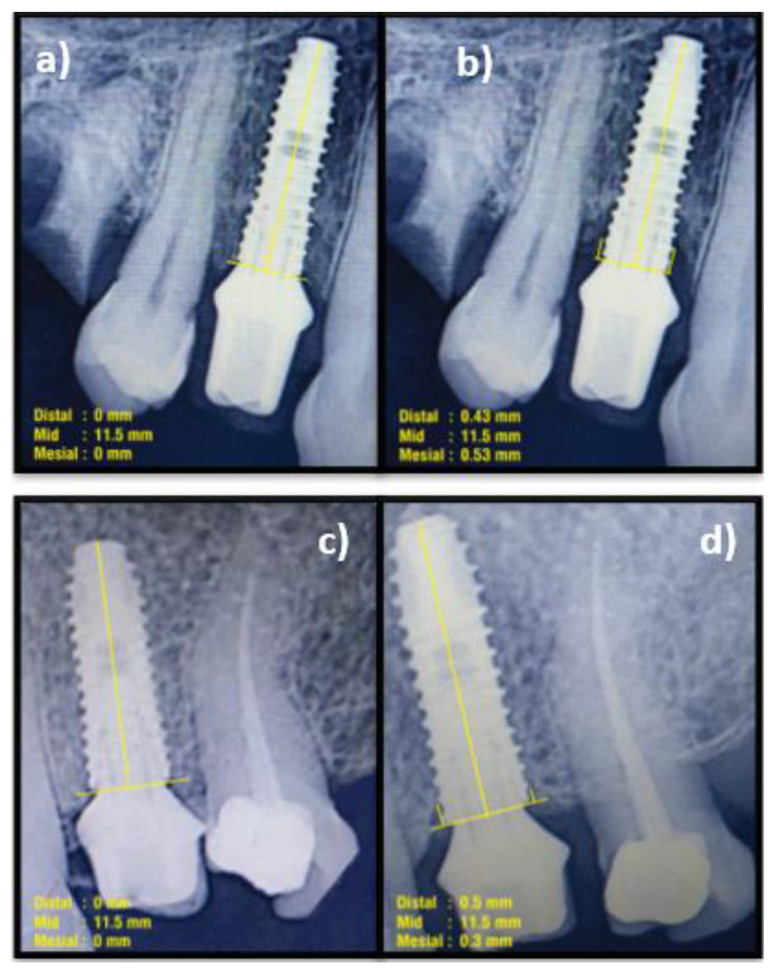
Measurement of marginal bone loss (MBL). (**a**) Immediately post-functional loading (Control); (**b**) 1-year post-functional loading (Control); (**c**) Immediately post-functional loading (test); (**d**) 1-year post-functional loading (Test).

**Table 1 biomimetics-07-00043-t001:** Intragroup comparison of clinical parameters between baseline, 6 months, and 1 year in control and test group.

Clinical Parameter	Timeline	Control	Test
Site-Specific Plaque Score (S-SPS) (%)	Between baseline to 6 months	0.31	0.00
	Between baseline to 1 year	0.15	−1.41
	Between 6 months to 1 year	0.31	−1.00
Site-Specific Bleeding Score (S-SBS) (%)	Between baseline to 6 months	0.31	0.00
	Between baseline to 1 year	0.15	−1.41
	Between 6 months to 1 year	0.31	−1.00
Peri-Implant Sulcus Depth (PISD) (mm)	Between baseline to 6 months	0.12	0.16
	Between baseline to 1 year	0.01 *	0.003 *
	Between 6 months to 1 year	0.01 *	0.003 *
Relative Position of Gingival Margin (PGM) (mm)	Between baseline to 6 months	0.08	0.03 *
	Between baseline to 1 year	0.01 *	0.01 *
	Between 6 months to 1 year	0.03 *	0.08

* *p* < 0.05 was considered as statistically significant.

**Table 2 biomimetics-07-00043-t002:** Intergroup comparison of clinical parameters at 6 months and 1 year.

Parameter	Timeline	Control	Test	*p*-Value
S-SPS	6 months	15.00 ± 12.91	12.50 ± 13.17	0.66
	1 year	17.50 ± 12.07	17.50 ± 12.07	1.00
S-SBS	6 months	15.00 ± 12.91	12.50 ± 13.17	0.66
	1 year	17.50 ± 12.07	17.50 ± 12.07	1.00
PISD	6 months	3.40 ± 0.24	3.42 ± 0.39	0.11
	1 year	3.23 ± 0.26	3.27 ± 0.34	0.52
PGM	6 months	2.90 ± 0.56	3.00 ± 0.66	0.85
	1 year	2.50 ± 0.52	2.70 ± 0.48	0.20

**Table 3 biomimetics-07-00043-t003:** Comparison of mean marginal bone loss (MBL) changes between control and test group after one year.

Parameter	Period (Year)	Group	Mean ± Sd	F-Value	*p*-Value
Mesial mean MBL (M-MBL) (mm)	1	Control	0.31 ± 0.11	0.01	0.89
		Test	0.35 ± 0.11		
Distal mean MBL (D-MBL) (mm)	1	Control	0.28 ± 0.09	1.90	0.18
		Test	0.33 ± 0.13		

MBL = Marginal bone loss.

**Table 4 biomimetics-07-00043-t004:** Pearson’s correlation of clinical and radiographic parameters at one year in the control and test group.

Clinical Parameters			Radiographic Parameter	
			Mesial Mean Crestal Bone Level (M-MCBL)	Distal Mean Crestal Bone Level (D-MCBL)
Site-specific Plaque score (S-SPS)	CONTROL	R-value	−0.19	−0.11
		*p*-value	0.59	0.74
	TEST	R-value	0.08	−0.003
		*p*-value	0.80	0.99
Site-specific Bleeding Score (S-SBS)	CONTROL	R-value	−0.19	−0.11
		*p*-value	0.59	0.74
	TEST	R-value	0.08	−0.003
		*p*-value	0.80	0.99
Peri Implant Sulcus Depth (PISD)	CONTROL	R-value	−0.56	−0.51
		*p*-value	0.09	0.12
	TEST	R-value	0.15	0.11
		*p*-value	0.67	0.74
Relative Position of Gingival Margin (PGM)	CONTROL	R-value	0.14	0.21
		*p*-value	0.69	0.54
	TEST	R-value	0.04	0.27
		*p*-value	0.89	0.44

## Data Availability

Not applicable.

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
