# Peer review of "Clinical and Radiological Outcomes for Guided Implant Placement in Sites Preserved with Bioactive Glass Bone Graft after Tooth Extraction: A Controlled Clinical Trial"

_biomimetics, 2022, doi:10.3390/biomimetics7020043_

Round 1

Reviewer 1 Report

Study title: Clinical and radiological outcomes for guided-implant placement in sites preserved with bioactive glass bone graft after tooth extraction: A controlled clinical trial

After reading the article, which I considered well structured and developed, it emerged some questions (below):

- why did the authors not develop a randomization?

- Could the authors include the sample calculation? (to justify n=20 implants)

- Could the authors include more specific features of the biomaterial used? (in order to understand better the resorption level and understand a little bit better the result)

- Review figure 1, there is some necessary adjusts to do.

I congratulate the authors for the article submitted.

Author Response

REVIEWER COMMENTS

Response to the reviewers comments and changes made

Academic Editor Comments

It would also help the reader if schematics are provided as to the procedures used pointing out the key aspects of the present approach.

Thanks for the suggestion.

As requested all the needed information has been put into a schematic representation format in Figure 1.

The manuscript is nicely presented. It would be useful to present a table comparing the pros and cons of different materials and methods.

All the necessary information on the materials and the GIS protocol has been adequately discussed in the manuscript section on materials and methods.

REVIEWER -1

why did the authors not develop a randomization?

Thanks for the query.

One group was augmented and the other group was non-augmented , which have been considered for implant placement using guided implant surgery. Further based on the objective of the study the groups were different –augmented and non-augmented sites but the treatment protocol was the same for both the groups and the effect of this treatment protocol  was observed so no randomisation  was followed.

Could the authors include the sample calculation? (to justify n=20 implants)

Thanks for the query.

Already the sample size calculation and the justification has been put in the materials and methods

Sample Size Calculation:

Sample size calculation was done based on results obtained from a study by Alessandro Pozzi et al  with 5% alpha error and 80% power based on Mean Bone Level measurements with 10 implants in each group with a total of 20 implants. Taking 20% drop out rate into consideration the sample size obtained is 24 edentulous sites which require implant placements with 12 implants to be placed in each group.

Could the authors include more specific features of the biomaterial used? (in order to understand better the resorption level and understand a little bit better the result)

Thanks for the recommendation , we have added the reviewers point of view in the introduction and highlighted in the manuscript.

Bioactive glass is a material of choice that has been shown to be more biocompatible as they contain hydroxyapatite, which has similar composition to that of bone and helps in the process of osteogenesis. The mechanism by which these bioactive glass forms bone is by causing an initial dissolution of the top most layers of the bioactive glass particulates which is followed by carbonated hydroxyapatite layer formation (hydroxycarbonate apatite, HCA) over the glass surface, and as a consequence elevated calcium ion levels and soluble silica are found in the area surrounding the dental implant. These soluble particles have the ability to form the bone matrix thereby resulting in the occurrence of the bioactivity, and the subsequent rate of deposition of the hydroxycarbonate apatite layer stimulates regeneration of bone.

Review figure 1, there is some necessary adjusts to do.

Corrections done as requested in the methodology section

REVIEWER -2

The study was not decreased in level but might be thought to do a randomization.

Thanks for the query.

One group was augmented and the other group was non-augmented , which have been considered for implant placement using guided implant surgery. Further based on the objective of the study the groups were different –augmented and non-augmented sites but the treatment protocol was the same for both the groups and the effect of this treatment protocol  was observed so no randomisation  was followed.

Fig. 1 - there is an error in the word "Excluded". Send another one, please.

Corrections done as requested in the methodology section

Reviewer 3:

The number of authors is extremely high. It almost reaches one author per implant placed. Given the relative lack of complexity of the study, the enormous number of authors is striking, so it should be reduced or adequately explained.

Respected Reviewer, Each of the authors who have been listed in the study have made significant contribution which has been listed in the author contribution segment. Kindly permit us to retain the same list.

The final epigraphs of the article are missing.

Thanks for the suggestion. We have added the epigraph in the conclusion as requested by the reviewer.

The article fails to identify the experiment itself. That is to say, what differentiates both groups is the performance or not of the graft after the extraction, but there is no reference to how this intervention was, what aspects were measured to ensure its success, as well as the success rates and complications of, I repeat , the experimental intervention. In this sense, the MBL or the clinical outcome of this intervention is a result of it. Possibly, the results of the graft and its follow-up were published in another previous article, which is not referenced, and this could be a salami publication. Or they have not been published, in which case they should be incorporated into this study.

The objective of the study: Evaluate, and compare the marginal bone loss (MBL) after 1-year implant placement using a guided implant surgical (GIS) protocol in grafted sockets and non-grafted sites

The inference of the study was to study the effectiveness of GIS protocol which can reduce the mean crestal bone loss which happens in an augmented site post 1 year implant placement.

We have discussed the GIS protocol effectively in our Discussion to speak for its effectiveness in the augmented sites.

I differ with the reviewers comment of this being a salami publication as this study as this study has not been reported to any other journal per se and I would like to bring it to the notice of the esteemed reviewer that the results which our study has brought about has given a new dimension in the use of GIS protocol in reducing mean crestal bone loss in augmented sites which we consider as novelty. The use of the bioactive bone graft in the augmented sites has mimicked regeneration which along with the guided protocol has given enhanced results.

Thank you 

Finally, this very clinical study does not seem to be useful in the journal Biomimetics, in addition to studying an aspect that has already been sufficiently discussed in the literature.

The use of a bioactive bone graft in socket preservation by itself mimics guided bone regeneration, which by itself is a quality for it to be represented in a bio mimetic journal.

Thank you

Reviewer 4: The paper "Clinical and radiological outcomes for guided-implant placement in sites preserved with bioactive glass bone graft after tooth extraction: A controlled clinical trial " is of some interest.

The abstract is clear; the introduction is quite exhaustive and concise, materials and methods are well described and with many pictures; the discussion is well conducted and highlights some limitations. Overall the study may be considered for publication in its current form

Thank you for your positive comments.

Reviewer 2 Report

This study, controlled clinical trial, is interesting and I considered well-developed. It evaluated 20 dental implants placed (MBL, SR and clinical aspects after 1 year: full-mouth plaque scores, bleeding on probing, site-specific plaque, site-specific bleeding scores, peri-implant sulcus depth), using guides and which received immediate functional loading; 10 sockets received graft and 10 non-received. The study was not decreased in level but might be thought to do a randomization.

- The study was registered in the Clinicaltrials.gov and approved by the ethical committee.

- Fig. 1 - there is an error in the word "Excluded". Send another one, please.

- Only one surgeon developed the surgeries, which avoid bias. And 2 different investigators were blinded (well done).

- I considered the results and discussion adequate (the authors included limitations of the study).

- Conclusion responded the main goal.

Author Response

Thank you for the comments and commendations.

- Fig. 1 - there is an error in the word "Excluded". Send another one, please.

R.: It was fixed.

- Only one surgeon developed the surgeries, which avoid bias. And 2 different investigators were blinded (well done). R.: Thank you.

- I considered the results and discussion adequate (the authors included limitations of the study). R.: We appreciated the comment

Reviewer 3 Report

The article is interesting although it has certain biases that must be corrected.
1.- The number of authors is extremely high. It almost reaches one author per implant placed. Given the relative lack of complexity of the study, the enormous number of authors is striking, so it should be reduced or adequately explained.
2.- The final epigraphs of the article are missing.
3.- The article fails to identify the experiment itself. That is to say, what differentiates both groups is the performance or not of the graft after the extraction, but there is no reference to how this intervention was, what aspects were measured to ensure its success, as well as the success rates and complications of, I repeat , the experimental intervention. In this sense, the MBL or the clinical outcome of this intervention is a result of it. Possibly, the results of the graft and its follow-up were published in another previous article, which is not referenced, and this could be a salami publication. Or they have not been published, in which case they should be incorporated into this study.
4.- Finally, this very clinical study does not seem to be useful in the journal Biomimetics, in addition to studying an aspect that has already been sufficiently discussed in the literature.

Author Response

(The authors gave the same response as above.)

Reviewer 4 Report

The paper "Clinical and radiological outcomes for guided-implant placement in sites preserved with bioactive glass bone graft after tooth extraction: A controlled clinical trial " is of some interest.

The abstract is clear; the introduction is quite exhaustive and concise, materials and methods are well described and with many pictures; the discussion is well conducted and highlights some limitations. Overall the study may be considered for publication in its current form

Author Response

Thank you very much for your revision and consideration.